# SpatialReasoner: Towards Explicit and Generalizable 3D Spatial Reasoning

**Wufei Ma*   Yu-Cheng Chou*   Qihao Liu***
**Xingrui Wang    Celso M de Melo[†]    Jianwen Xie[o]    Alan Yuille**

Johns Hopkins University, [†]DEVCOM Army Research Laboratory, [o] Lambda Inc

## Abstract

Despite recent advances on multi-modal models, 3D spatial reasoning remains a challenging task for state-of-the-art open-source and proprietary models. Recent studies explore data-driven approaches and achieve enhanced spatial reasoning performance by fine-tuning models on 3D-related visual question-answering data. However, these methods typically perform spatial reasoning in an implicit manner and often fail on questions that are trivial to humans, even with long chain-of-thought reasoning. In this work, we introduce SpatialReasoner, a novel large vision-language model (LVLM) that addresses 3D spatial reasoning with explicit 3D representations shared between multiple stages–3D perception, computation, and reasoning. Explicit 3D representations provide a coherent interface that supports advanced 3D spatial reasoning and improves the generalization ability to novel question types. Furthermore, by analyzing the explicit 3D representations in multi-step reasoning traces of SpatialReasoner, we study the factual errors and identify key shortcomings of current LVLMs. Results show that our SpatialReasoner achieves improved performance on a variety of spatial reasoning benchmarks, outperforming Gemini 2.0 by 9.2% on 3DSRBench, and generalizes better when evaluating on novel 3D spatial reasoning questions. Our study bridges the 3D parsing capabilities of prior visual foundation models with the powerful reasoning abilities of large language models, opening new directions for 3D spatial reasoning. Our project page is available here.

## 1 Introduction

3D spatial reasoning studies how models perceive, understand, and reason about 3D object properties and spatial relationships. It is not only a fundamental task for vision-language models to achieve human-level intelligence, but also crucial to a range of downstream applications in robotics [14, 18] and embodied AI [10]. Despite the recent advancements of large multi-modal models, such as GPT-4o and Qwen2.5-VL, their 3D spatial reasoning capabilities remain limited and fall far behind human-level performance [9, 31, 51]. Recent 3D-aware large vision-language models (LVLMs) injected 3D knowledge by fine-tuning the model on synthetic 3D-related question-answer pairs [9, 33] and achieved improved performance on spatial reasoning benchmarks [31, 45, 51]. On the other hand, large proprietary models such as Gemini 2.0 [15] have advanced 3D parsing by directly predicting 3D object bounding boxes, enabling the development of powerful generalist robotics models [43].

We identify two key challenges in 3D spatial reasoning: (1) *3D thinking*–the ability to decompose a complex 3D spatial reasoning question into small, manageable steps, and (2) *3D computation*–the ability to solve these thinking steps in a consistent and accurate manner. As shown in Figure 1, prior reasoning methods adopt long chain-of-thought reasoning to tackle the problem but do not

---

*Equal first author

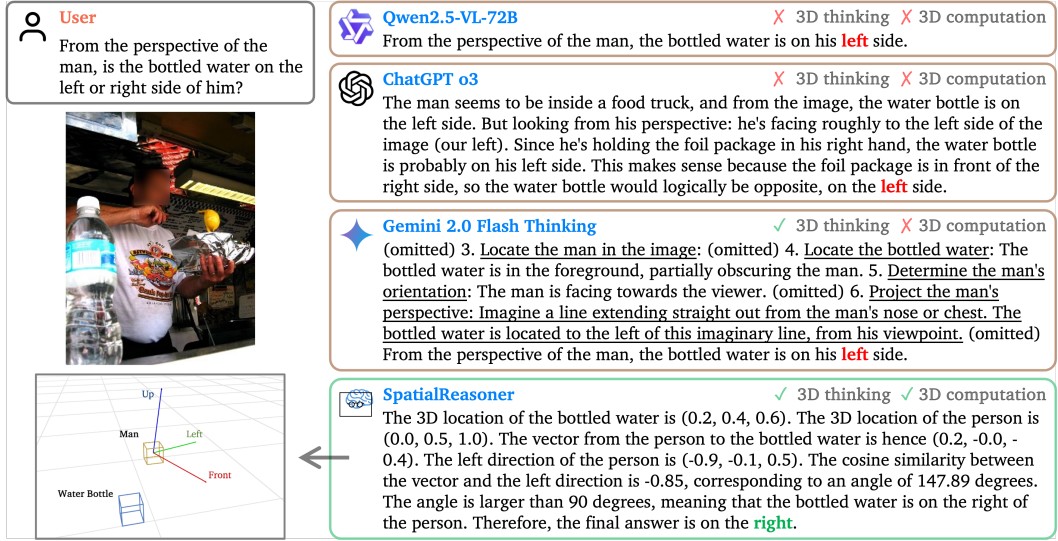

Figure 1: **Comparing 3D spatial reasoning of our SpatialReasoner with previous state-of-the-art models.** Our SpatialReasoner builds on explicit 3D representations, performs 3D computation, and reasons about the final answer. Although Gemini 2.0 can also break down complex 3D spatial reasoning questions into small and tractable steps, it lacks reliable 3D computation and the long chain-of-thought reasoning ultimately leads to a wrong answer.

have explicit 3D computation. ChatGPT o3 [36] fails to adopt a systematic approach to solving the problem and instead relies on other visual cues to assist the reasoning (*e.g.*, location of the foil package). In contrast, the Gemini 2.0 thinking model [15] employs an organized strategy to tackle the problem but ultimately fails to arrive at the correct answer due to a lack of reliable 3D computation.

In this work, we present SpatialReasoner, a novel large vision-language model (LVLM) built with (1) explicit 3D representations and (2) enhanced and generalizable 3D thinking. Specifically, our SpatialReasoner adopts explicit 3D representations, such as 3D locations and orientations, as an interface that enables coherent and reliable reasoning across multiple stages, *i.e.*, 3D perception, computation, and reasoning. On the other hand, we would like to learn enhanced 3D thinking capabilities that generalize to novel question types not seen during training. Hence we explore a two-stage training strategy following prior works [17]. In Stage I, we apply supervised fine-tuning (SFT) to equip the LVLM with explicit 3D representations, enhancing 3D perception and computation capabilities of the model. Then in Stage II, we leverage reinforcement learning (RL) to develop robust and generalizable 3D thinking built on explicit 3D representations.

Inspired by previous 3D pseudo-annotation pipelines in [9, 11, 33], we synthesize basic 3D perception, *i.e.*, detection and pose estimation, and 3D computation question-answering data interleaved with explicit 3D representations. We further generate standard spatial reasoning question-answer pairs with chain-of-thought reasoning that breaks down complex 3D spatial reasoning questions into multiple steps–3D perception, computation, and reasoning. Experimental results demonstrate that our SpatialReasoner with explicit 3D representations can significantly enhance 3D spatial reasoning abilities of LVLMs and generalize to novel question types.

Besides enhancing 3D spatial reasoning capabilities of LVLMs, reasoning with explicit 3D representations allows us to interpret the reasoning process and to study the failure modes of LVLMs. We find that the accuracy of 3D perception lags significantly behind that of 3D computation, suggesting that most errors in downstream VQA tasks still stem from failures in 3D perception. Moreover, by predicting key 3D information, such as 3D object locations and orientations, as intermediate results, our SpatialReasoner enables compositional reasoning for 3D spatial tasks [46]. This not only allows our method to generalize better to novel spatial reasoning questions, but also makes it easily extensible to other tasks that build on our explicit 3D representations.

Besides improving spatial reasoning performance on a variety of benchmarks, we experiment on various LVLMs fine-tuned with different combinations of data and training methods to study the key

factors toward improved 3D spatial reasoning. Our empirical results lead to the following insights: (1) For 3D-aware VLMs, SFT offers a more scalable approach than RL that requires high-quality 3D-aware data, which is often difficult to obtain. Our recipe of RL followed by SFT achieves the best overall performance; (2) 3D-aware LVLMs fine-tuned with RL generalize better than SFT when tested on novel 3D spatial reasoning questions, echoing prior findings [12]; (3) Standard LVLMs often exploit 2D reasoning as a shortcut to tackle 3D spatial reasoning problems, whereas our SpatialReasoner avoids the spurious correlations and always reasons with explicit 3D representations, achieving improved and robust performance on challenging real-world 3D spatial reasoning datasets.

In summary, our contributions are as follows: (1) We introduce SpatialReasoner, a novel LVLM that solves 3D spatial reasoning problems with a compositional approach based on explicit 3D representations. (2) Reasoning with 3D representations allows us to interpret the reasoning process and to study the failure modes of LVLMs. (3) Our SpatialReasoner effectively improves the 3D spatial reasoning performance on a range of benchmarks and extensive experimental results provide valuable findings on data and training strategy designs for future development of 3D-aware LVLMs.

## 2 Related Works

**3D Spatial reasoning.** 3D spatial reasoning explores how models perceive and reason about 3D object properties and relationships. Early works [46, 47, 48] studied this in simulated environments or datasets with object-level annotations [20, 25, 7, 11, 45, 33]. More recent efforts construct benchmarks based on real-world imagery [31, 51] and improve model performance via synthetic QA data [9, 11, 33, 39]. However, these models often rely on implicit reasoning, offering little interpretability or intermediate 3D computation. Moreover, it remains unclear whether the 3D spatial reasoning capabilities acquired from data-driven fine-tuning can generalize to novel question types that require complex 3D computation over different combinations of 3D perception outputs.

**Explicit 3D representations.** Explicit 3D representations simplify reasoning and expose model failures. Simulation-based works [35] used neural-symbolic methods to parse object-level structures, while PO3D [46] and its extensions [47, 48] showed that structured visual modules enable interpretable reasoning [32]. These results in simulation systems reveal the limitations of current LVLMs and highlight the importance of the visual module for a successful reasoning model.

**Post-training.** Post-training aligns pre-trained models with downstream objectives via SFT [49, 54] and RL [5, 6, 37, 41], improving formatting and reward-guided alignment [21, 44]. Recent systems like GPT-4 [1], Claude-3.5 [2], and DeepSeek-R1 [17] showcase these techniques. Despite recent advances, post-trained models still struggle with generalization under distribution shifts and novel tasks. While SFT stabilizes outputs but often overfits, RL improves adaptability [12], and combining both shows promise [17], making post-training vital for robust, aligned LLMs.

**Test-time scaling.** Scaling inference-time compute improves performance without retraining. Approaches include beam search [30, 16], best-of-N sampling [42], and MCTS [13], alongside prompting methods like CoT [50] and ToT [53]. Strategically allocated test-time compute has proven effective for enhancing reasoning in both unimodal and multimodal models. Building on this, we apply CoT reasoning to 3D spatial reasoning by fine-tuning models to generate step-by-step rationales, providing both accurate and interpretable results.

## 3 SpatialReasoner

### 3.1 Overview

In this section, we introduce our SpatialReasoner for explicit and generalizable 3D spatial reasoning. Our SpatialReasoner features two key designs: (1) explicit 3D representations that serves as interface to support multi-stage spatial reasoning, *i.e.*, 3D parsing, computation, and reasoning (see Figure 2), and (2) generalizable spatial reasoning from multi-stage training (see Figure 4).

In Section 3.2, we present the explicit 3D representations and describe how our model is trained to predict and to interpret the 3D representations for spatial reasoning. Then in Section 3.3 we discuss our training strategies, exploring standard supervised fine-tuning, reinforcement learning, as well as

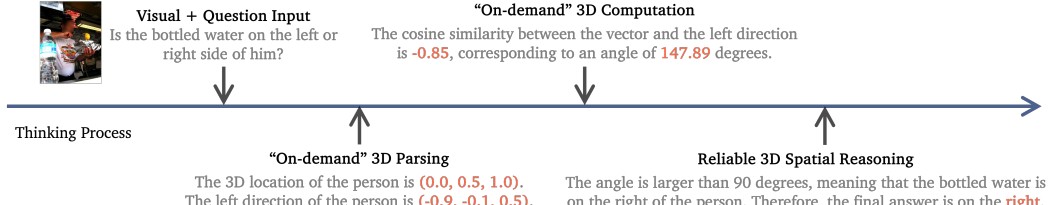

**Visual + Question Input**
Is the bottled water on the left or right side of him?

**"On-demand" 3D Computation**
The cosine similarity between the vector and the left direction is **-0.85**, corresponding to an angle of **147.89** degrees.

Thinking Process

**"On-demand" 3D Parsing**
The 3D location of the person is **(0.0, 0.5, 1.0)**.
The left direction of the person is **(-0.9, -0.1, 0.5)**.

**Reliable 3D Spatial Reasoning**
The angle is larger than 90 degrees, meaning that the bottled water is on the right of the person. Therefore, the final answer is on the **right**.

Figure 2: **Overview of our SpatialReasoner design.** Our SpatialReasoner adopts explicit 3D representations as an interface to enable coherent and reliable multi-stage spatial reasoning, *i.e.*, 3D parsing, computation, and reasoning.

3D-aware process rewards. Lastly we introduce our 3D-aware data generation pipeline and different variants of training data used at different stages in Section 3.4.

## 3.2 Learning Explicit 3D Representations

Despite improved spatial reasoning abilities achieved by 3D-aware VLMs, such as SpatialRGPT [11] and SpatialLLM [33], and advanced proprietary models like Gemini 2.0 [15], these methods lack explicit 3D representations and rely on natural language to perform 3D spatial reasoning. For example, Gemini 2.0 describes object poses as "facing towards the viewer and slightly to its right" and estimates 3D distances with phrases such as "far behind some other object". Such natural language descriptions are inefficient and often not accurate enough for complex 3D spatial reasoning.

Therefore, we propose integrating LVLMs with explicit 3D representations, such as 3D locations and poses, to serve as an accurate and reliable interface shared across stages of 3D spatial reasoning (see Figure 2). Our SpatialReasoner can predict explicit 3D representations as intermediate results or take them as inputs to perform basic 3D computations or complex spatial reasoning tasks.

**Explicit 3D representations.** We define explicit 3D representations in a calibrated camera 3D space, which is a standard camera 3D space calibrated with the extrinsics of the camera. As illustrated in Figure 3, the calibrated camera 3D space has its $z$-axis aligned with the $z$-axis of the 3D world space, and the origin on the $z$-axis is positioned close to the ground plane. Although estimating 3D object locations is easier in original camera 3D space, adopting explicit 3D representations in calibrated camera 3D space offers many advantages for subsequent spatial reasoning: (1) the $z$-coordinates directly correspond to object heights, (2) estimating 3D spatial relationships such as "above" and "below" is largely simplified, and (3) objects often are on a plane parallel to the ground plane, which reduces many 3D spatial relationships to simpler 2D problems.

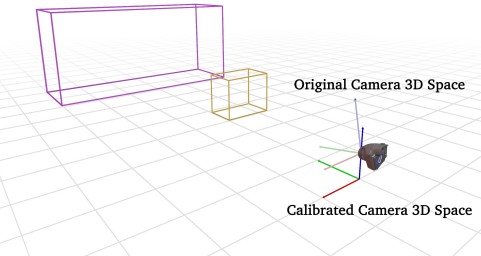

Original Camera 3D Space

Calibrated Camera 3D Space

Figure 3: **Comparison between original and calibrated camera 3D space.** Our explicit 3D representations are defined within calibrated camera 3D space that simplifies subsequent 3D computation and reasoning.

**A unified interface for explicit 3D spatial reasoning.** Explicit 3D representations serve as an interface enabling coherent and accurate 3D spatial reasoning across stages (see Figure 2). For 3D perception, the model predicts object locations and orientations as 3D vectors, then estimates explicit distances or angles based on these predictions. Finally, the model aggregates explicit 3D information from earlier stages to reason about and answer 3D spatial questions.

## 3.3 Training Strategies

**Supervised fine-tuning (SFT).** We adopt a two-stage post-training strategy to equip the model with explicit, generalizable 3D spatial reasoning capabilities. In the first stage, SFT serves as a critical initialization step, aligning the pre-trained LVLM with curated 3D-annotated datasets. By

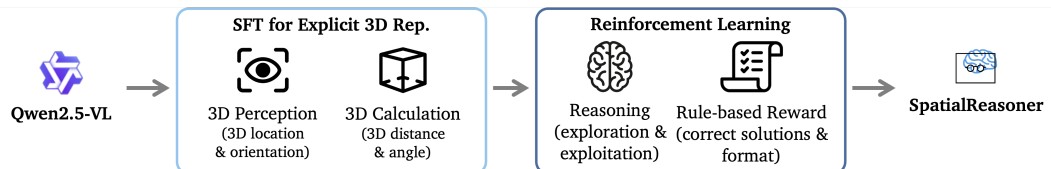

Figure 4: **Overview of our SpatialReasoner training.** We adopt a multi-stage training strategy: In Stage I, we apply supervised fine-tuning (SFT) to equip the LVLM with explicit 3D representations; then in Stage II, we leverage reinforcement learning (RL) to develop robust and generalizable 3D spatial reasoning built on explicit 3D representations.

optimizing a maximum likelihood objective over paired input-output sequences, the model learns to identify objects in 3D scenes and predict explicit 3D representations such as object locations and orientations. This structured supervision enables the model to expose interpretable intermediate reasoning traces during spatial computations—*e.g.*, calculating relative distances and angles. However, due to its reliance on static demonstrations, SFT shows limited capacity to generalize beyond observed reasoning patterns. As shown in Section 4.3, SFT-trained models tend to memorize spatial templates in the training set and struggle with novel compositions or combinatorial variations in 3D queries.

**Reinforcement learning (RL).** To overcome this limitation, we further post-train the SFT-initialized model using reinforcement learning (RL) with rule-based rewards. Treating the spatial reasoning task as a sequential decision process, each reasoning step is framed as an action within a Markov Decision Process (MDP), and policy gradients are optimized using GRPO [41]. Importantly, we integrate a reward scheme that provides structured reward signals reflecting both the correctness of final answers and the coherence of intermediate 3D computation steps. This enables multi-turn optimization where the model learns to revise inaccurate inferences and explore alternative reasoning paths. Empirically, this RL stage substantially improves generalization to out-of-distribution 3D spatial questions (Table 3), particularly in settings involving multi-object arrangements. The combined SFT+RL strategy therefore balances response formatting and perceptual grounding with adaptive reasoning and robustness, making it especially suited for complex 3D spatial understanding.

**Reward and policy optimization design.** To train SpatialReasoner via reinforcement learning, we design a composite reward scheme capturing both answer correctness and reasoning quality. For outcome rewards, we use an accuracy reward—aligned with the multiple-choice evaluation metric from MMBench [28]—which gives positive feedback only when the model selects the correct answer, and a format reward [17] that encourages structured, readable outputs. We also explore process rewards to assess whether models can learn structured reasoning without SFT. These include a reasoning steps reward, which promotes use of structured indicators (*e.g.*, "First", "Next"), and a 3D-aware process reward that checks for necessary spatial terms like distance or orientation. The reward signal is then calculated through the accuracy of the presence of each term. Though not used in the final model, these rewards serve as diagnostics for emergent reasoning behavior. Finally, we ablate the KL divergence term in GRPO and find that it harms training accuracy despite stabilizing output lengths (Section 4.3). We thus remove it in our final setting.

### 3.4 Training Data

To enable LVLMs to predict and reason with explicit 3D representations and to post-train LVLMs to solve various challenging 3D spatial reasoning questions, we generate a series of 3D-aware training data. We extend the data generation pipeline in [9, 11, 33]. Our process begins with generating 3D pseudo-annotations, followed by optional human verification, and ends with constructing various VQAs and chain-of-thought reasoning based on the 3D pseudo-annotations.

**Pseudo 3D ground-truths.** We extend the 3D pseudo-annotation pipeline proposed in [9, 33] and generate various 3D annotations, such as object category, 3D location, and 3D pose, on unlabeled images from the OpenImages dataset [22]. Based on the object-level 3D annotations, we then apply rule-based methods to derive ground-truth labels for a range of 3D spatial relationships. Despite significant progress in visual foundation models for segmentation [38], metric depth estimation [52], and object pose estimation [34], we notice many factual errors in the generated 3D ground-truth,

| Method | Mean | Height | Location | Orientation | Multi-Object |
|---|---|---|---|---|---|
| ***Open-Sourced Generalist*** | | | | | |
| LLaVA-v1.5-7B [26] | 38.1 | 39.1 | 46.9 | 28.7 | 34.7 |
| LLaVA-Next-8B [24] | 48.4 | 50.6 | 59.9 | 36.1 | 43.4 |
| Cambrian-1-8B [45] | 42.2 | 23.2 | 53.9 | 35.9 | 41.9 |
| Qwen2.5-VL-3B-Instruct [4] | 43.9 | 45.2 | 56.8 | 35.7 | 35.7 |
| Qwen2.5-VL-7B-Instruct [4] | 48.4 | 44.1 | 62.7 | 40.6 | 40.5 |
| Qwen2.5-VL-72B-Instruct [4] | 54.9 | 53.3 | 71.0 | 43.1 | 46.6 |
| ***Open-Sourced Specialist*** | | | | | |
| SpaceLLaVA [40] | 42.0 | 49.3 | 54.4 | 27.6 | 35.4 |
| SpatialBot [8] | 41.0 | 40.4 | 54.4 | 31.9 | 33.5 |
| SpatialLLM [33] | 44.8 | 45.8 | 61.6 | 30.0 | 36.7 |
| SpatialRGPT [11] | 32.7 | **55.9** | 39.0 | 27.8 | 20.0 |
| SpatialRGPT *w/* depth [11] | 48.4 | 55.9 | 60.0 | 34.2 | 42.3 |
| ***Proprietary*** | | | | | |
| GPT-4o-mini | 39.7 | 44.3 | 52.4 | 21.0 | 36.5 |
| GPT-4o | 44.2 | 53.2 | 59.6 | 21.6 | 39.0 |
| Claude 3.5 V Sonnet | 48.2 | 53.5 | 63.1 | 31.4 | 41.3 |
| Gemini 2.0 Flash | 49.8 | 49.7 | 68.9 | 32.2 | 41.5 |
| Gemini 2.0 Flash (thinking) | 51.1 | 53.0 | 67.1 | 35.8 | 43.6 |
| QwenVLMax | 52.0 | 45.1 | 70.7 | 37.7 | 44.8 |
| ***Ours*** | | | | | |
| SpatialReasoner-Zero | 54.0 | 46.4 | 67.3 | 48.4 | 47.2 |
| SpatialReasoner-SFT | 58.3 | 51.9 | 73.5 | 50.7 | 50.3 |
| SpatialReasoner | **60.3** | 52.5 | **75.2** | **55.2** | **51.8** |

Table 1: **Comparison with previous state-of-the-art methods on 3DSRBench [31].** Our SpatialReasoner outperforms previous open-source and proprietary methods on challenging 3D spatial reasoning problems in 3DSRBench.

particularly when mistakes (*e.g.*, missing objects or inaccurate object poses) propagate through the data pipeline. Therefore, we adopt a series of aggressive filtering steps to ensure the quality of our training data, including: (1) removing images with densely cluttered scenes, (2) excluding object categories that are difficult to segment or estimate pose for, and (3) discarding boundary cases that could lead to ambiguity.

**Human verification.** Despite leveraging state-of-the-art visual foundation models in our data generation pipeline and applying multiple filtering strategies, the 3D pseudo-annotations remain susceptible to factual errors. Specifically, small issues such as missing objects or inaccurate 3D pose predictions propagate through later stages of the pipeline, leading to factual errors in the final spatial relationship pseudo-annotations. To assess the impact of data quality, we create a smaller but higher-quality dataset by manually verifying the correctness of the 3D pseudo-annotations.

**Training data variants.** Based on the obtained 3D ground-truth, we can generate different variants of data for fine-tuning. Specifically, we consider the following: (1) *Basic3D-QA* consists of basic 3D perception and 3D computation question-answering data. This can be used to learn explicit 3D representations without training on various 3D spatial relationships considered in downstream tasks. (2) *SR-QA* contains visual question-answering pairs about various 3D spatial relationships, following previous 3D-aware datasets [11, 33]. (3) *SR-CoT* extends SR-QA and comprises chain-of-thought reasoning with explicit 3D representations. Questions are answered in a step-by-step manner.

## 4 Results

### 4.1 Experimental Setup

**Baselines.** We compare our SpatialReasoner with the following three types of baseline models. (1) *Open-sourced generalists*: such as LLaVA [26], Cambrian-1 [45], and Qwen2.5 [4] that are trained on

| Method | CV-Bench-3D | | GQA | | | | | |
|---|---|---|---|---|---|---|---|---|
| | Depth | Distance | Mean | Choose | Compare | Logical | Query | Verify |
| Qwen2.5-VL-7B-Instruct | 82.5 | **83.2** | 58.8 | 82.7 | 71.5 | 78.9 | 40.6 | 82.5 |
| SpatialReasoner-Zero | 77.5 | 81.8 | 60.2 | 81.6 | 67.9 | 79.2 | 43.8 | 82.1 |
| SpatialReasoner-SFT | 85.2 | 71.5 | **62.0** | 82.8 | 72.2 | **81.4** | **45.5** | **83.2** |
| SpatialReasoner | **87.3** | 73.3 | 61.8 | **83.2** | **81.1** | 71.8 | 45.2 | 82.9 |

Table 2: **Performance on CV-Bench-3D and GQA.** Our SpatialReasoner also improves the spatial reasoning performance on GQA [19] and depth-related questions in CVBench-3D [45]. For distance-related questions, unlike Qwen2.5 that exhibits excessive dependence on 2D shortcuts, our SpatialReasoner employs rigid 3D spatial reasoning and achieves compelling performance. Regarding the performance on distance-related questions, see Section 4.2 and Section C for detailed discussions.

| Method | Mean | Height | In-Distribution | | Novel |
| | | | Location | Orientation | Multi-Object |
|---|---|---|---|---|---|
| Qwen2.5-VL-7B-Instruct | 48.4 | 44.1 | 62.7 | 40.6 | 40.5 |
| SpatialReasoner-Zero | 53.7 | 40.6 | 68.4 | 50.2 | **46.6** |
| SpatialReasoner-SFT | 52.2 | 44.9 | 69.5 | 48.9 | 40.0 |
| SpatialReasoner | **56.4** | **52.5** | **72.6** | **54.1** | 43.4 |

Table 3: **Evaluation of generalization ability by finetuning on simpler (in-distribution) 3D spatial reasoning questions and evaluating on complex (novel) questions types unseen during training.** Our SpatialReasoner-Zero and SpatialReasoner demonstrates superior zero-shot generalization, indicating RL fosters more robust and flexible reasoning than SFT.

general vision-language data. (2) *Open-sourced specialists*: We evaluate SpaceLLaVA [40] (a public re-implementation of SpatialVLM [9]), SpatialBot [8] that enhances fine-grained spatial reasoning and robot control with RGB and depth images, and SpatialLLM [33] that fine-tunes a LLaVA model with multi-stage 3D-informed training. Note for fair comparison, we evaluate SpatialBot with RGB inputs only. (3) *Proprietary models* such as GPT-4o [1] and Claude 3.5 [3] that were trained on abundant web-scale data and for Gemini 2.0 [15], additional 3D-aware post-training.

**Evaluation benchmarks.** We evaluate spatial reasoning abilities of various models on three spatial reasoning benchmarks. *3DSRBench* [31] is a comprehensive 3D spatial reasoning benchmark with 2,100 questions and studies various 3D awareness and reasoning abilities with a robust evaluation setup. *CVBench* [45] is a vision-centric benchmark that assesses models at classic vision tasks with a range of 2D and 3D understanding VQAs. In this work, we focus exclusively on 3D-related questions, *i.e.*, CVBench-3D, as 2D left-right relationships can lead to ambiguity with 3D left-right questions considered in 3DSRBench. *GQA* [19] is a widely adopted benchmark that studies visual reasoning and compositional question answering on a range of spatial relationships between objects.

## 4.2 Advancing 3D Spatial Reasoning

We evaluate SpatialReasoner on 3DSRBench [31], CVBench-3D [45], and GQA [19] to assess 3D perception, computation, and reasoning. As shown in Table 1, SpatialReasoner achieves a new state-of-the-art 60.3% mean accuracy on 3DSRBench, outperforming Gemini 2.0 Flash (49.8%) and Claude 3.5 Sonnet (48.2%). It shows notable improvements on Location (+6.3%) and Orientation (+14.6%) questions, and achieves 51.8% (+8.2%) on Multi-Object reasoning relative to the second-best model, indicating stronger 3D perception and complex spatial understanding.

On CVBench-3D and GQA (Table 2), SpatialReasoner achieves 87.3% on depth-related questions in CVBench-3D, and improved performance on GQA's Compare category, further supporting its multi-object reasoning capability. These gains validate our multi-stage training: SFT equips the model with explicit 3D representations, while RL enhances adaptive reasoning and generalization.

Despite this, we observe a performance drop on CVBench-3D distance questions, in contrast to substantial gains on similar 3DSRBench cases ("multi-object-closer-to", 34.3% → 70.9%). We attribute

| Method | Mean |
|---|---|
| Qwen2.5-VL-7B-Instruct | 48.4 |
| SR-SFT | 58.3 |
| SR-SFT (*w/o* exp. 3D rep.) | 51.9 |

Table 4: **Comparisons between SpatialReasoner with and without explicit 3D representations and CoT reasoning.** Results highlight the benefits of explicit 3D representations as an interface to support enhanced 3D spatial reasoning.

| Method | Mean |
|---|---|
| Qwen2.5-VL-7B-Instruct | 48.4 |
| SpatialReasoner-SFT | 58.3 |
| SpatialReasoner-SFT (+HQ SFT) | 54.7 |
| SpatialReasoner-Zero | 54.0 |
| SpatialReasoner-Zero (*w/* KL) | 52.4 |
| SpatialReasoner-Zero (*w/* 3D Rwd) | 54.6 |

Table 5: **Ablation study on various design choices in RL and SFT.** Notably with 3D-aware rewards, SpatialReasoner-Zero produces coherent chain-of-thought reasoning on explicit 3D representations and improves benchmark performance.

this gap to shortcut-driven 2D biases in CVBench-3D. Unlike existing LVLMs, SpatialReasoner relies on explicit 3D reasoning, offering greater robustness to such spurious cues (Section C).

### 4.3 Analyses and Findings

**Generalization abilities.** While SFT stabilizes outputs, it often overfits training distributions, limiting generalization to out-of-distribution (OOD) variations [12]. In contrast, outcome-based RL promotes transferable reasoning strategies and perceptual adaptability. To examine this, we compare SpatialReasoner-SFT (SFT-only), SpatialReasoner-Zero (RL-only), and SpatialReasoner (SFT+RL) on multi-object reasoning, with all multi-object data removed during training.

As shown in Table 3, SpatialReasoner-Zero achieves 46.6%, outperforming SpatialReasoner-SFT (40.0%) and SpatialReasoner (43.4%), confirming RL's superior zero-shot generalization ability. Although SFT+RL improves over SFT, it does not fully match RL-only performance, indicating that RL fosters more flexible reasoning than memorization-prone SFT.

When multi-object training data are available (Table 1), SFT attains 50.3% via pattern matching, but drops to 40.0% when data are withheld, whereas RL-based models retain stable performance (46.6% vs. 47.2%). Mean accuracy further supports this: SpatialReasoner-Zero maintains performance (53.7% vs. 54.0%), while SpatialReasoner-SFT degrades significantly (58.3% to 52.2%), reinforcing that RL enables more robust generalization.

**Compositional reasoning.** We assess whether structured 3D reasoning can emerge from process rewards alone, without curated CoT. As shown in Figure 8, SpatialReasoner-Zero trained with 3D-aware rewards generates coherent multi-step reasoning grounded in 3D cues, improving interpretability over the baseline relying on shallow shortcuts. This demonstrates the model's ability to synthesize novel compositional reasoning strategies from basic 3D perception and computation skills. Quantitatively, it achieves a performance gain from 54.0% to 54.6% (Table 5), indicating that outcome-driven reward shaping can induce coherent reasoning without human-crafted CoT. Nonetheless, for the final SpatialReasoner, we use SFT to guide reasoning traces, as it outperforms process rewards alone.

**Scaling of training computation.** We ablate on the scaling of training computation for SpatialReasoner-SFT and SpatialReasoner in Figure 5. Results show that with more training steps, SpatialReasoner-SFT starts to overfit and exhibit decreased performance, while SpatialReasoner trained with RL retains a stable and competitive performance.

**Scaling of training data.** 3D-related tasks often lack high-quality data with 3D (pseudo-)annotations due to the cost and expertise required. While synthetic or pseudo-labeled data is more scalable, they introduce noise and domain gaps. To study data scaling, we mix 1.2K verified and 24K unverified samples and compare SpatialReasoner with and without RL. As shown in Figure 6, RL performs best with only the 1.2K verified data, whereas SpatialReasoner-SFT benefits from more unverified data. This suggests that SFT is more tolerant to noisy pseudo-annotations, while RL favors fewer but higher-quality samples.

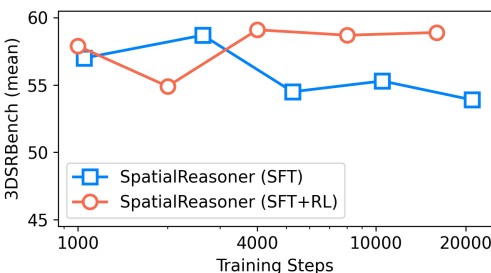
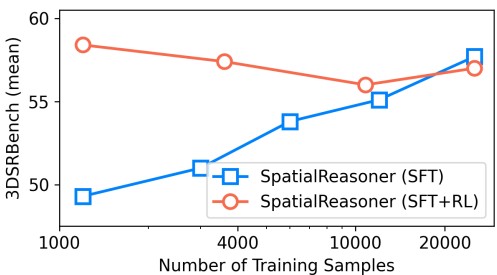

Figure 5: **Scaling of training computation.** SpatialReasoner-SFT may overfit to training data while SpatialReasoner trained with RL retains a stable and competitive performance.

Figure 6: **Scaling of training data by mixing 1.2K human-verified with 24K unverified data.** Results show SFT remains a compelling choice when scaling with potentially noisy training data.

**Ablation study on explicit 3D representations.** To study the importance of explicit 3D representations and step-by-step 3D computations, we train a variant of SpatialReasoner on the same data but without explicit 3D representations. Results in Table 4 show that explicit 3D representations can enhance 3D spatial reasoning by a wide margin.

**Ablation study on KL divergence.** We evaluate KL divergence regularization in GRPO and find it harms learning in our setting. As shown in Figure 9, adding KL causes training to collapse, while removing it yields stable improvements. The shorter completions without KL may reflect mitigated length bias [29], improving token efficiency. As shown in Table 5, SpatialReasoner-Zero reaches 54.0% accuracy without KL vs. 52.4% with it. We thus omit KL for better spatial reasoning.

**Ablation Study on SFT+RL vs. SFT+SFT** To verify the performance gains are not merely from additional data exposure, we compare sequential SFT+RL against SFT+SFT. From Table 5, while SpatialReasoner-SFT achieves 58.3% accuracy, applying a second round of SFT (SFT+SFT) degrades performance to 54.7%. In contrast, RL after SFT (SFT+RL) improves accuracy to 60.3%. These results indicate repeated SFT exacerbates overfitting to training patterns, whereas RL effectively builds on the structured outputs from SFT to promote adaptive and generalizable 3D spatial reasoning.

## 4.4 Interpreting Failure Modes

Explicit 3D representations enhance both spatial reasoning and interpretability [46]. We categorize 3D spatial reasoning into two stages: 3D perception (parsing key 3D information from the image input) and 3D reasoning (computing 3D metrics and deriving final answer). To quantitatively analyze model behavior, we manually verify 300 questions with human-verified 3D answers. For 3D perception, we evaluate questions on object location and orientation; for 3D reasoning, models estimate distances, depths, and angles from 3D inputs, with accuracy (within $\pi/6$) for angles and mean error for positions. As shown in Table 7, models reason more accurately than they perceive–indicating most VQA errors stem from flawed 3D perception, consistent with our qualitative visualizations. Additionally, RL slightly degrades perception and reasoning accuracy, likely due to rewards focusing only on final answers and reasoning format, not intermediate 3D outputs.

## 5 Conclusions

In this paper, we presented SpatialReasoner that performs explicit and generalizable 3D spatial reasoning by predicting intermediate 3D representations across perception, computation, and reasoning stages. Our two-stage post-training pipeline with SFT followed by RL achieves state-of-the-art results on multiple benchmarks while generalizing well to novel tasks. By analyzing model behavior, we show that explicit 3D reasoning improves both accuracy and interpretability, and reveals key limitations in 3D perception. This work highlights the value of structured reasoning and adaptive training for robust 3D understanding, paving the way for future research in multimodal spatial reasoning.

**Acknowledgements.** W.M. and A.Y. acknowledges support from the Office of Naval Research with N00014-23-1-2641 and ARL award W911NF2320008.

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

# A  Limitations

Our system has a few limitations. First, despite the use of reinforcement learning (RL), supervised finetuning data with chain-of-thought reasoning and explicit 3D representations remains crucial as a warm-up. Training with RL alone–*i.e.*, SpatialReasoner-Zero–fails to produce high-quality reasoning processes. Second, SpatialReasoner consistently performs explicit multi-step 3D reasoning, regardless of the difficulty of the problems. Ideally the model should adopt a hybrid strategy–leveraging commen sense and visual cues for simpler questions while reserving explicit 3D computation and reasoning for more challenging problems. This would retain many strong and generalizable knowledge from the base Qwen2.5-VL model while improving inference efficiency.

# B  Implementation Details

We train SpatialReasoner using different combinations of curated datasets and training objectives. Starting from the Qwen2.5-VL-7B [4] base model, we first apply SFT with 24k curated SR-CoT data alongside 24k randomly sampled LLaVA [27] data, resulting in SpatialReasoner-SFT. Next, we further train SpatialReasoner-SFT with RL using 1.2k SR-QA examples, leading to our final SpatialReasoner. For comparison, if we instead fine-tune SpatialReasoner-SFT with SFT on the same 1.2k SR-QA set, we obtain SpatialReasoner-SFT (+HQ SFT), serving as an ablation study baseline. In parallel, we directly fine-tune the base model with RL using the 1.2k SR-QA data without prior SFT, resulting in SpatialReasoner-Zero. Additionally, to investigate if explicit 3D perception ability can enable the model to self-organize reasoning trajectories under process rewards, we train the base model with SFT using 12k Basic3D-QA data alongside 12k randomly sampled LLaVA data before applying RL training on the 1.2k SR-QA data, yielding SpatialReasoner-Zero (*w/* 3D Rwd).

We conduct all training experiments using 4×NVIDIA H100 80GB HBM3 GPUs. For SFT, we train the model for 10 epochs (approximately 20K steps with a batch size of 6) on the combined 24k SR-CoT and 24k LLaVA datasets. For RL training, we train for 100 epochs (approximately 13K steps with a batch size of 12) on the 1.2k SR-QA dataset, using 1 GPU with vLLM [23] for efficient inference acceleration. For the experiments withholding multi-object training examples, we double the number of training epochs to compensate for the reduced training set size. We set the learning rate to 5e-6 for SFT and 5e-7 for RL, both following a cosine learning rate scheduler with a warm-up ratio of 0.1. In the KL divergence ablation study, we set the KL penalty weight to 0.04. We monitored the model every 1K training steps and reported results based on the best-performing checkpoint.

# C  2D Reasoning as a Shortcut

As shown in Table 1 and Table 2, our SpatialReasoner outperforms previous open-source and proprietary models on 3DSRBench [31], and achieves notable improvements on GQA [19] (from 58.8% to 61.8%) and depth-related questions in CVBench-3D [45] (from 82.5% to 87.3%). However, if we focus on multi-object 3D distance-related questions in CVBench-3D [45] and 3DSRBench [31], we observe contradictory results: SpatialReasoner achieves a substantial improvement of 21.5% on 3DSRBench, but exhibits a notable performance drop of 9.9% on CVBench-3D (see Table 6).

**We attribute this discrepancy to the abundant shortcuts in distance-related questions in CVBench-3D.** From the qualitative examples in Figure 7, the provided 2D bounding boxes in CVBench-3D can be exploited as shortcuts to answer the 3D spatial reasoning question. Rather than reasoning about 3D distances between objects, we can easily derive the correct answer by comparing the 2D distances between the red and blue boxes and between the red and green boxes. Meanwhile, 3DSRBench is a human-collected VQA dataset and manually avoid such spurious correlation, *e.g.*, "objects closer in 3D space are also closer in 2D image plane". For the 3DSRBench example in Figure 7, the bounding box of the dog is actually closer to the bounding box of the man in black that farther away in 3D space.

Given the 2D bounding box annotations in CVBench-3D and 3DSRBench, we derive a simple heuristic to answer distance-related spatial reasoning questions by simply comparing L2 distances between 2D centers of object bounding boxes. We achieve an 80.2% accuracy on distance questions in CVBench-3D and 34.3% in 3DSRBench. **This demonstrates that baseline models such as Qwen2.5-VL are largely exploiting 2D spatial reasoning as a shortcut to answer complex 3D spatial**

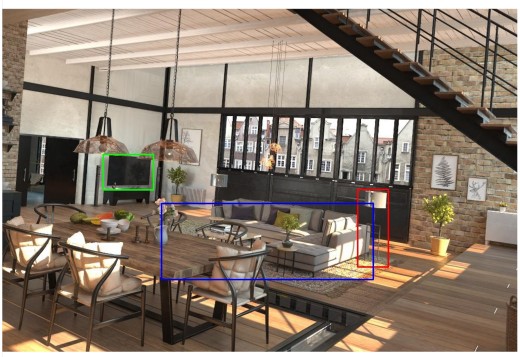

| CVBench-3D (Distance) | 3DSRBench (Multi-Object Closer to) |
| --- | --- |
| | (bounding box added for visualization) |

Question: Estimate the real-world distances between objects in this image. Which object is closer to the lamp (highlighted by a red box), the sofa (highlighted by a blue box) or the television (highlighted by a green box)?

Answer: Sofa.

Question: Consider the real-world 3D locations of the objects. Which is closer to the dog, the person in red or the person in black?

Answer: Person in red.

Figure 7: Comparison between (multi-object) distance-related questions in CVBench-3D [45] and 3DSRBench [31].

| Method | CVBench3D Distance | 3DSRBench multi-object-closer-to |
| --- | --- | --- |
| 2D Heuristic | 80.2 | 34.3 |
| Qwen2.5-VL-7B-Instruct [4] | 83.2 | 49.4 |
| SpatialReasoner | 73.3 | 70.9 |

Table 6: Comparison between Qwen2.5-VL [4] and SpatialReasoner on (multi-object) 3D distance-related questions in CVBench-3D [45] and 3DSRBench [31].

**reasoning questions.** Meanwhile, with our 3D-aware post-training, our SpatialReasoner adopts explicit 3D representation for 3D spatial reasoning questions. **The trade-off between exploiting 2D shortcuts and adopting explicit 3D representations results in slightly lower performance of SpatialReasoner on test data with abundant spurious correlations, but more importantly, leads to a robust and largely improved performance on a challenging real-world dataset.**

> 💡 **Visual cues or explicit 3D reasoning? LVLMs that exploit 2D reasoning as shortcuts may achieve improved performance on test data with abundant spurious correlations (*e.g.*, distance questions in CVBench-3D). However, they cannot genuinely solve 3D spatial reasoning problems and fall far behind SpatialReasoner that builds on explicit 3D spatial reasoning when tested on challenging real-world problems in 3DSRBench.**

## D    Additional Results

We present the following additional results:

1. **Table 7.** We study the failures modes of SpatialReasoner. Results show that 3D computation is more accurate than 3D perception, indicating that most VQA errors stem from flawed 3D perception.

2. **Figure 8.** Qualitative comparisons between different variants of SpatialReasoner.

3. **Figure 9.** Ablation study results on KL divergence.

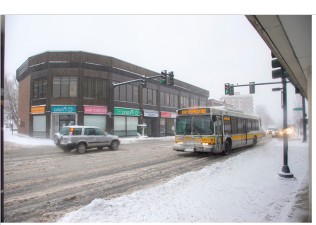

**User**
Consider the real-world 3D locations and orientations of the objects. Which side of the bus is facing the green board? A. right, B. front, C. left, D. back.

**SpatialReasoner-SFT *w/o* explicit 3D representation**
D. back.

**SpatialReasoner-Zero**
<think>The green board is on the front of the bus.</think><answer>B.</answer>

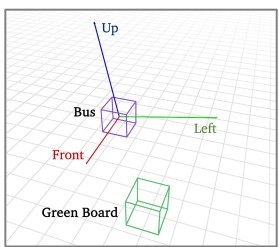

**SpatialReaonser-Zero *w/* 3D-aware process reward**
<think>First, I will determine the front direction of the bus. The front direction of the bus is (-0.17, 0.25, -0.96). Next, I will calculate the 3D location of the bus and the green board. The 3D location of the bus is (3.97, 2.35, 17.15), and the 3D location of the green board is (1.75, 1.65, 13.95). Then, I will calculate the vector from the bus to the green board. The vector is (-1.70, 0.00, 0.99). After that, I will determine the left direction of the bus. The left direction of the bus is (-0.92, 0.37, -0.29). Finally, I will check the angle between the front direction and the green board and compare it with the angle between the left direction and the green board. The angle between the front direction and the green board is 0.29 and the angle between the left direction and the green board is 0.33. Therefore, the correct answer is the front side of the bus facing the green board.</think><answer>B. front</answer>

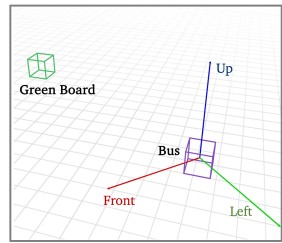

**SpatialReaonser-SFT**
The 3D location of the bus is (-4.3, 1.9, 17.7). The 3D location of the green board is (4.9, 2.9, 22.1). The vector from the bus to the green board is hence (9.2, 0.9, 4.4). The left direction of the bus is (-0.7, 0.0, -0.7). The cosine similarity between the vector pointing to the green board and the left direction is -0.94, corresponding to an angle of 160.87 degrees. Thus the angle between the vector pointing to the green board and the right direction is 19.13 degrees. The front direction of the bus is (0.7, 0.0, -0.7). The cosine similarity between the vector pointing to the green board and the front direction is 0.34, corresponding to an angle of 70.30 degrees. Thus the angle between the vector pointing to the green board and the back direction is 109.70 degrees. Among the four directions, the smallest angle is right direction, with an angle of 19.13 degrees. Thus right side of the bus is facing the green board. Therefore, the final answer is A. right.

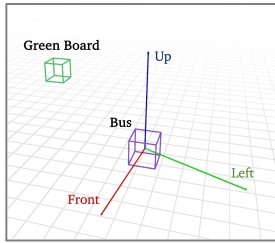

**SpatialReaonser**
<think>The 3D location of the bus is (-3.7, 2.1, 14.7). The 3D location of the green board is (2.2, 3.1, 19.8). The vector from the bus to the green board is hence (5.9, 1.0, 5.1). The left direction of the bus is (-0.9, 0.0, -0.4). The cosine of the angle between the vector pointing to the green board and the left direction is -0.99, corresponding to an angle of 172.09 degrees. Thus the angle between the vector pointing to the green board and the right direction is 7.91 degrees. The front direction of the bus is (0.1, 0.0, -1.0). The cosine of the angle between the vector pointing to the green board and the front direction is -0.15, corresponding to an angle of 98.47 degrees. Thus the angle between the vector pointing to the green board and the back direction is 81.53 degrees. Among the four angles, the smallest is the right direction (7.91 degrees), which is the answer to this question.</think><answer>right</answer>

Figure 8: **Qualitative comparisons.** Our explicit 3D spatial reasoning improves interpretability over the baseline that relies on shallow shortcuts.

# E  Open Access

All code, data, and models will be available on our project page to support reproducibility and benefit the research community.

1. Codebase for our full 3D-aware data generation pipeline.
2. Codebase for our SFT and RL finetuning.
3. Synthesized 3D-aware training data.
4. Weights of our SpatialReasonerand SpatialReasoner-SFT.

|        |        | 3D Perception |        | 3D Reasoning |        |        |
|--------|--------|---------------|--------|--------------|--------|--------|
| Method | | Orientation ($\uparrow$) | Location ($\downarrow$) | Angle ($\uparrow$) | Distance ($\downarrow$) | Depth ($\downarrow$) |
| SpatialReasoner-SFT | | 35.5 | 0.91 | 55.0 | 0.17 | 0.13 |
| SpatialReasoner | | 31.0 | 1.05 | 52.5 | 0.19 | 0.25 |

Table 7: **Studying failure modes of SpatialReasoner.** We observe that 3D reasoning can estimate angles, distances, and depths a lot more accurately than 3D perception that predicts orientations and locations from visual features.

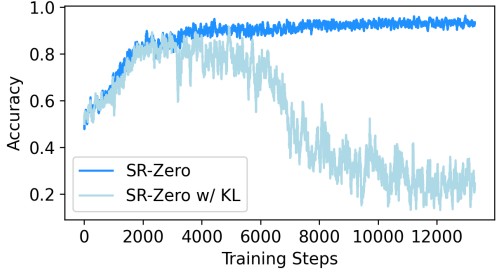

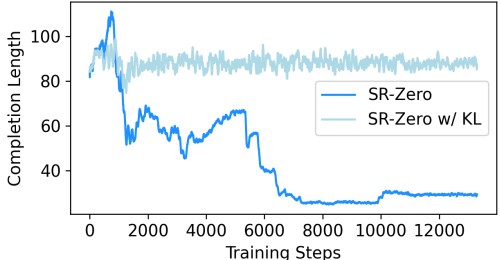

(a) Accuracy of SpatialReasoner-Zero *w/* and *w/o* KL divergence.

(b) Completion length of SpatialReasoner-Zero *w/* and *w/o* KL divergence.

Figure 9: **Training curve between SpatialReasoner-Zero and SpatialReasoner-Zero *w/* KL.**

