# OpenReview forum: "SpatialReasoner: Towards Explicit and Generalizable 3D Spatial Reasoning"
_NeurIPS.cc/2025/Conference — NeurIPS 2025 poster_

### Official Review · Reviewer_X79Q · 2025-06-03

**Clarity:** 3
**Significance:** 3
**Originality:** 4
**Rating:** 5
**Confidence:** 3

**Summary:**

The paper introduces SpatialReasoner for 3D spatial reasoning. SpatialReasoner employs structured 3D representations. The model understands the space with transformations among different space coordinates. The model is trained in two stages: SFT and RL to enhance adaptability and generalization. Experimental results across multiple benchmarks show that the model achieves excellent performance.

**Questions:**

1. How would the model perform on robotics setting? Please consider some spatial understanding datasets on robots:

[1] Song, Chan Hee, et al. "Robospatial: Teaching spatial understanding to 2d and 3d vision-language models for robotics." Proceedings of the Computer Vision and Pattern Recognition Conference. 2025.
[2] Li, Yun, et al. "Sti-bench: Are mllms ready for precise spatial-temporal world understanding?." arXiv preprint arXiv:2503.23765 (2025).

I don't have further questions. The paper is good to be accepted to NeurIPS.

**Ethical Concerns:**

["NO or VERY MINOR ethics concerns only"]

**Final Justification:**

I recommend the paper to be accepted.

**Limitations:**

Yes, the authors have discussed the limilations of their method.

**Quality:**

3

**Strengths And Weaknesses:**

Strength:

1. The work is solid with a generated 3D-aware training data. Also, SFT and RL are both adopted and analyzed.

2. Comprehensive experiments are done on various baseline models and benchmarks.

3. The paper is well written and easy to follow

Weakness:
1. Please provide more visual examples of your dataset, and comparasion of model outputs.

2. What is the accuracy of estimation 3D positions? (On-demand 3D parsing in Figure 2)

3. The images in the article does not appear to be vector images.

4. Please provide more metadata. e.g. the size of training dataset. For line 207~208, the authors may want to provide more details.

---

> ### Author Rebuttal · Authors · 2025-07-31
>
> We thank the reviewer for the time and efforts to review our paper. We appreciate the positive feedback, mentioning that our paper is "well-written and easy to follow", our work "solid", and our experiments "comprehensive". We discuss new results and address detailed concerns below. We will incorporate all valuable feedback and suggestions in our future revision.
>
> **More visual examples of training data and comparison of model outputs.** We thank the reviewer for the helpful suggestion. We show some qualitative examples in Figure 1 and Figure 8. However, due to NeurIPS policy restrictions, we cannot upload images or provide new qualitative results at this stage. We will include additional visual examples in the revised version of the paper.
>
> **Accuracy of estimation 3D positions.** As SpatialReasoner predicts 3D locations (3D center of the objects without height/width/length) in the learned 3D coordinate system (Figure 3), we cannot directly evaluate it on standard object detection datasets. To study how well our SpatialReasoner model predicts 3D locations and orientations, we (1) present qualiative examples for visual inspection (Figure 1 and Figure 8), and (2) evaluate our model on unseen testing data with human verified 3D groundtruths (Table 7). Results in Table 7 show that
> 1. 3D reasoning (estimating distance and angle from location and direction) is more accurate than 3D perception (estimating location and estimation from image).
> 2. After RL, we notice a small drop in both 3D perception and computation abilities. For future work, we will consider more training designs to ensure reliable 3D perception and computation, while improving the overall MLLM reasoning abilities.
>
> While we cannot directly compare with other expert models on 3D object detection, results in Table 3 provide valuable insights that guide the design of model, data, and training strategy.
>
> **Concern regarding image quality.** We appreciate the reviewer’s comment regarding image quality. Due to time constraints prior to the submission deadline, we included some JPEG images in the manuscript to meet the file size limitations. In our revision, we will replace these images with vector-format illustrations and ensure appropriate DPI settings to improve presentation quality.
>
> **More metadata on size of the training data.** The training data of our SpatialReasoner contains (1) 24k visual-question-answering data with chain-of-thought reasoning interleaved with 3D pseudo-labels (e.g., locations and orientations), and (2) an additional 1.2k visual-question-answering data where the answers are manually checked for better quality. We will include more dataset statistics in our revision.
>
> **Regarding human inspection on L207-208.** While our training data is built on predictions from state-of-the-art visual foundation models such as Segment Anything and Depth Anything, our training data is still error-prone. Specifically, we found that the estimated 3D orientations are often less accurate than 3D locations, and errors from one visual foundation model can be propagated and accumulated in later stages, leading to degraded data quality. Hence we adopt a human verification process to study the influence of data quality in SFT and RL post training. We will discuss more details on our human verification process in the supplementary materials of our revision, including the following: (1) web-based UI for human labeling, (2) statistics of human verification, and (3) percentage of raw VQA data filtered out by human annotators for each question type.
>
> **How would the model perform on robotics setting.** We thank the reviewer for the suggestion. As our SpatialReasoner is an image-based model, we cannot directly evaluate on STI-Bench. However, we evaluate our SpatialReasoner on RoboSpatial for reference.
>
> Since the training data of RoboSpatial is not released, we evaluate zero-shot performance of our SpatialReasoner on RobotSpatial-Home and compare with several baselines in the RoboSpatial paper. Results show that our SpatialReasoner performs on-par with prior works. Moreover, we note the following:
> 1. Our SpatialReasoner attempts to solve various problems in RoboSpaital with explicit 3D representations (see example below). This demonstrates that our SpatialReasoner learns strong and transferable 3D knowledge.
> 2. Although post-trained with various spatial reasoning VQA data, SpatialReasoner does not clearly outperforms previous methods on RoboSpatial. This is likely because RoboSpatial includes a range of 2D ego-centric left/right spatial relationships in the image plane, while our SpatialReasoner are only finetuned with 3D object-centric left/right spatial reasoning data. For future work we will consider various data mixing approaches and improve the general spatial reasoning abilities.
>
> Lastly we will involve a paragraph in our revision that discusses the broader impact of our SpatialReasoner in the area of robotics, including the results on RoboSpatial, how spatial reasoning abilities of SpatialReasoner can benefit robotics tasks, and how explicit 3D representations can be extended in robotics settings.
>
> **Example output of SpatialReasoner on RoboSpatial.**
> * **Question:** Can the dish soap fit right of the bowl? Answer yes or no.
> * **Ground truth answer:** No.
> * **SpatialReasoner:** \<think\> The 3D location of the dish soap is (-0.6, 0.5, 1.1). The 3D location of the bowl is (-0.9, 0.8, 1.7). The vector from the bowl to the dish soap is hence (0.3, -0.3, -0.6). The length of this vector is 0.79. The size of the bowl is roughly 0.27. The size of the dish soap is roughly 0.44. The ratio of the distance between the two objects and the object sizes is 1.75. The answer is no. \</think\>\<answer\>no\</answer\>
>
> **Quantitative results on RoboSpatial.** Note that for "context" questions model predictions must follow certain formats so models without finetuning on the unreleased RoboSpaital training data performs very badly on this split.
>
> | model | Configuration | Context | Compatibility |
> |:-|:-:|:-:|:-:|
> | SpaceLLaVA | 61.0 | 2.5 | 61.0 |
> | 3D-LLM | 39.8 | 0.0 | 35.2 |
> | Molmo | 58.6 | 0.1 | 18.1 |
> | SpatialReasoner | 61.0 | 0.0 | 60.6 |

---

> > ### Comment · Reviewer_X79Q · 2025-08-01
> >
> > Thank you for your rebuttal. My concerns have been addressed. I recommend the papar to be accepted to NeurIPS.

---

> > > ### Author Response · Authors · 2025-08-03
> > > **Response to Reviewer X79Q**
> > >
> > > Thanks once again for your insightful comments and helpful suggestions. We will carefully address and incorporate them into our revision.

---

> > > ### Author Response · Authors · 2025-08-05
> > > **Response to Reviewer X79Q**
> > >
> > > Dear Reviewer X79Q, thank you again for the positive evaluation and constructive feedback. As we approach the end of the reviewer-author discussion period, we would like to kindly check if there are any remaining questions or concerns you would like us to address. If our rebuttal has fully addressed your concerns, we would be grateful if you might consider increasing your score to reflect a strong accept.

---

### Official Review · Reviewer_rCGV · 2025-06-28

**Clarity:** 1
**Significance:** 2
**Originality:** 3
**Rating:** 3
**Confidence:** 4

**Summary:**

This paper explores the integration of explicit 3D representations into the outputs of large vision-language models (LVLMs). The motivation is that 3D representations offer interpretable and debuggable structures, enabling clearer reasoning and computation over visual content. The proposed approach demonstrates performance improvements on both in-domain objects and, to a lesser extent, on out-of-domain objects.

**Questions:**

- **Dataset**: What's the size of the SFT and RL dataset? Any numbers showing the distribution of different question types?

- **Results**
  - SpatialReasoner-Zero outperforms on novel objects. Could this suggest that explicit 3D representations may actually hurt performance in some cases? For instance, requiring the model to generate 3D outputs might increase hallucinations.
  - In L248, what does “multi-object-closer-to” mean?

- **Training**: Line 179: It’s unclear how the 3D-aware process reward is defined or explored. More detail would help.

**Ethical Concerns:**

["NO or VERY MINOR ethics concerns only"]

**Final Justification:**

See my response to the rebuttal.

**Limitations:**

The paper does not explicitly discuss its limitations. Given that its core contribution is the integration of explicit 3D representations into reasoning traces and the proposal of a data synthesis method, the lack of clarity around the data construction process is a significant issue. The writing should be improved to make the data synthesis pipeline and its characteristics more transparent.

**Paper Formatting Concerns:**

I don't find any format concerns.

**Quality:**

2

**Strengths And Weaknesses:**

# Strengths

- The use of explicit 3D representations in LVLM outputs is a novel and compelling idea
- Comprehensive evaluation across both in-domain and out-of-domain objects


# Weakness
- Section 3.4 lacks sufficient detail on data synthesis. It's unclear how the dataset is constructed, its size, or the distribution of question types, making reproducibility difficult.
- The method for computing the 3D coordinates in Figure 2 is not clearly explained. It's also unclear what units are used and whether the representation can handle distant objects, such as cars in street scenes.
- The presentation of experimental results is difficult to follow. Key findings are scattered, and understanding the full picture often requires digging into the appendix. For example, the analysis in Table 6 on spurious correlations in CVBench-3D depth is insightful, but easy to overlook.

---

> ### Author Rebuttal · Authors · 2025-07-30
>
> We thank the reviewer for the time and effort to review our paper. We appreciate the positive feedback, mentioning that our method of using explicit 3D representations "novel and compelling" and our evaluation "comprehensive". We discuss new results and address detailed concerns below. We will incorporate all valuable feedback and suggestions in our future revision.
>
> **Implementation details on data synthesis.** We thank the reviewer for raising this concern. We will include more technical details of our data synthesis pipeline in our revision. For reproducibility, we will open-source code, data, and models to (1) reproduce our data generation process with 3D pseudo-labels and question-answering, and (2) reproduce key experimental results in our paper. Our code can be reused to benefit future research or be extended for other studies. We summarize key technical aspects as follows:
>
> To post-train multi-modal LLMs for better spatial understanding, previous works explored various visual foundation models such as Segment Anything and Depth Anything, and generated pseudo-labels for 3D depths, distances, and orientations [9,11,33] (L187-194). However, we extend beyond previous spatial VQA data synthesis pipelines from the following two aspects:
> * Spatial relationships and reasoning trajectory: Compared to [9,11], which only studies object 3D depths and distances or [33] that studies a limited number of orientation relationships, we study a broader range of spatial relationships that are both challenging and require multi-step spatial reasoning. Moreover, answers in our generated spatial VQA data also contain chain-of-thought reasoning trajectories interleaved with explicit 3D locations and orientations, which are found to benefit models’ spatial reasoning.
> * Data quality: as we extend to more complex spatial relationships, we notice an increasing number of factual errors due to the long multi-step synthesis pipeline. Hence, we adopt a series of aggressive filtering based on model predictions and heuristics (L197-202), which is followed by an optional human verification stage. This leads to a smaller (24k vs over 1M) but much higher quality spatial VQA dataset. We further study the influence of data quality on SFT and RL in L278-284.
>
> **Unit of 3D coordinates in Figure 2 and its applicability.** Our SpatialReasoner learns to predict these 3D coordinates from supervised finetuning (SFT) on our generated data. The pseudo-labels for 3D locations are obtained with metric depth estimation (L196), so the units are in meters. Regarding method applicability, our training data is built on web images from the OpenImages dataset, which involves a wide range of scenes and objects. Illustrations in Figure 8 demonstrate that our approach with explicit 3D representations also applies to street scenes with distant objects such as buses and shop boards. We will involve more qualitative examples in our revision, as NeurIPS does not allow image uploads during the rebuttal stage.
>
> **Presentation of experimental results is difficult to follow.** We thank the reviewer for pointing out this concern. Our SpatialReasoner includes a range of design components, such as data generation (different size and quality), reasoning trajectories (with and without 3D representations), and training methods (SFT, RL, 3D-aware process rewards). We conducted experiments to study the influence of each component, aiming to demonstrate the strengths of our approach as well as identify the weaknesses. Some results in the appendix are referenced (e.g., spurious cues in Section C from L250), but we agree with the reviewer that the clarity and organization of the experimental section need to be improved. We will reorganize our experimental results in the following four sections and add a summary paragraph before going through all the results.
> 1. Standard benchmark results: Table 1 and Table 2.
> 2. Ablation study: Table 4 and Table 5.
> 3. Failure case analyses: Table 7 and spurious cues in Section C.
> 4. Scaling results and design considerations: Figure 5 and Figure 6.
>
> **Training data statistics.** The training data of our SpatialReasoner includes: (1) 24k spatial VQA with pseudo 3D labels generated by our data synthesis pipeline, (2) another 1.2k spatial VQA synthesized and filtered by human verifiers, and (3) 24k standard VQA randomly sampled from the mix 665k SFT data used in LLaVA v1.5. For SFT, we jointly finetune SpatialReasoner on a combination of (1) + (2) + (3). For RL, we finetune the model on (2) only. As the training data are synthesized by our data generation pipeline, our spatial VQA training data are uniformly distributed across different question types.
>
> **SpatialReasoner-Zero outperforms on novel objects.** For clarification, all SpatialReasoner models (SFT, RL, Zero) involve exploiting 3D representations, and the difference is how they obtain the reasoning trajectories (see Figure 8). The goal of Table 3 is to study how post-trained spatial MLLMs generalize to novel spatial questions, which is an important yet largely understudied problem.
>
> Specifically, in Table 3, we report the generalization results of various MLLMs to novel question types. Models are finetuned on questions about height, location, and orientations, and we explore if the learned spatial knowledge generalizes to novel question types about spatial relationships between multiple objects ($\geq$ 3). From the results in Table 3, we conclude the following:
> 1. From the comparison between "Qwen2.5-VL-7B-Instruct" and "SpatialReasoner", we find that our SpatialReasoner learns explicit 3D representations, such as 3D locations and orientations, that transfer to novel question types and benefit spatial reasoning.
> 2. From the comparison between "SpatialReasoner" and "SpatialReasoner-Zero", we find that while SFT on synthesized spatial VQA data can largely benefit the in-distribution spatial reasoning abilities, it falls behind our SpatialReasoner-Zero model trained with 3D-aware process rewards. Without reliance on SFT data with crafted chain-of-thought reasoning, SpatialReasoner-Zero learns to explore spatial reasoning trajectories from the process reward functions and generalizes better to novel question types compared to the SFT counterpart (L257-260).
>
> We will include more discussions on these results in our revision.
>
> **Clarification on "multi-object-closer-to".** "Multi-object-closer-to" is one of the 12 question types defined in 3DSRBench. We discuss it in L248 as it is the question type that is most similar to the questions in CV-Bench distance-related questions. For example,
> * Multi-object-closer-to in 3DSRBench: Consider the real-world 3D locations of the objects. Which is closer to the desk, the bed or the fridge?
> * Distance-related questions in CVBench-3D: Estimate the real-world distances between objects in this image. Which object is closer to the chair (highlighted by a red box), the refrigerator (highlighted by a blue box) or the door (highlighted by a green box)?
>
> **Clarification on the 3D-aware process reward.** We apologize for the confusion and will introduce the motivation and detailed designs of our 3D-aware process rewards more clearly in our revision. We summarize the key technical aspects below.
>
> In this work, we consider three types of reward functions: accuracy reward, format reward, and 3D-aware process reward.
> * The accuracy and format rewards follow the DeepSeek-R1 design that checks if the predicted final answer matches the ground truth and if the format of the answer follows a reasoning step \<think\>...\</think\> and an answering step \<answer\>...\</answer\>.
> * Process rewards have been considered in [R1,R2,R3]. We propose novel 3D-aware process rewards to encourage the model to predict necessary 3D representations in the reasoning step. This is realized by implementing a range of regex patterns that recognize spatial representations --- 3D locations, distances, front/left direction, angles, etc. Then, given the question types, we reward the model if the reasoning trajectory contains relevant spatial representations. As an example, for a multi-object distance question, we give reward=1 if the reasoning trajectory contains 3D locations and distances, and reward=0 if the reasoning trajectory contains 3D orientations and directions.
>
> | reward | value | example |
> |:-|:-:|:-|
> | accuracy | 1 | \<answer\>red sofa\</answer\> (*matches the ground truth*) |
> | | 0 | \<answer\>blue sofa\</answer\> (*does not match the ground truth*) |
> | process | 1 | \<think\>The 3D location of the red sofa is…\</think\> \<answer\>red sofa\</answer\> |
> | | 0 | \<think\>The 3D location of the red sofa is… red sofa\</answer\> |
> | 3D-aware process | 1 | \<think\>The 3D location of the red sofa is…\</think\> |
> | | 0 | \<think\>The red sofa is next to the dining table…\</think\> |
>
> **Limitations of our work.** We discussed limitations of our work in the NeurIPS submission checklist (L575) and in Section A of our supplementary materials — including the reliance on supervised finetuning data and the inference efficiency of predicting nuanced 3D representations. We apologize for the confusion regarding the data construction process and will clarify the writing in our revision. Moreover, we will open-source all code to reproduce training data generation and to support future research.
>
> [R1] Scaling LLM Test-Time Compute Optimally can be More Effective than Scaling Model Parameters
>
> [R2] Math-Shepherd: Verify and Reinforce LLMs Step-by-step without Human Annotations
>
> [R3] VisualPRM: An Effective Process Reward Model for Multimodal Reasoning

---

> > ### Author Response · Authors · 2025-08-05
> > **Response to Reviewer rCGV**
> >
> > Dear Reviewer rCGV, thank you for your time and effort to review our paper. We appreciate the constructive feedback and will integrate them in our revision. As we approach the end of the reviewer–author discussion period, we would like to kindly check if our rebuttal has addressed your concerns or if there are any follow-up questions we can clarify. If our rebuttal has fully resolved your concerns, we would be grateful if you might consider increasing your rating.

---

> > ### Comment · Reviewer_rCGV · 2025-08-05
> > **Response to rebuttal**
> >
> > I appreciate the authors’ detailed rebuttal, which clarified my concerns regarding the data synthesis process. However, I remain skeptical about certain part of the experimental results and conclusions. The results appear fragmented and lack a strong, consistent signal. Overall, my rating to the submission is still around borderline. **I will keep my score, but happy to engage in the further discussion with AC.**
> >
> > The followings are my justification:
> > 1. Skeptical about SpatialReasoner-Zero: Given the simplicity of the reward design, it is difficult to believe that the model is truly learning to produce “good-enough” 3D coordinates to answer the questions. Generating accurate 3D coordinates is inherently a very challenging task, yet the current reward design lacks any explicit term that would encourage the model to generate meaningful 3D representations. My guess is that the model may simply produce arbitrary coordinates to exploit the reward signal, without integrating these coordinates into its final reasoning process.
> > 2. While the paper presents many experiments, they are coupled with strong claims (e.g., L260) that are not sufficiently supported, e.g., as a reader, I am still not sure how well the model use the 3D coordinates. I value numerical results, but I also consider it crucial to understand whether the model is truly reasoning through its generated 3D coordinates.
> >
> > Overall, I find the idea of explicitly generating 3D coordinates for spatial reasoning promising, and the data generation aspect is clearer after the rebuttal. However, the experimental evidence still sends sparse and noisy messages, and the rebuttal does not fully convince me that the model is genuinely learning meaningful 3D coordinate representations.

---

> > > ### Author Response · Authors · 2025-08-06
> > > **Response to Reviewer rCGV**
> > >
> > > We thank the reviewer for the continued engagement and thoughtful reflections. We appreciate the reviewer’s insights but we would like to make the following clarifications.
> > >
> > > **About the training of SpatialReasoner-Zero.** We agree with the reviewer that 3D-aware process rewards alone do not lead to good and reliable 3D coordinates. In fact, for the model to learn native spatial reasoning trajectories on top of meaningful 3D coordinates, we first finetune the base model on Basic3D-QA data that provides 3D coordinate supervision but no spatial reasoning (see implementation details on L501-504). Then with the learned meaningful 3D representations, we demonstrate that SpatialReasoner-Zero can acquire native spatial reasoning trajectories from 3D-aware process awards. We could also see from Figure 8 that SpatialReasoner-Zero does not contain any meaningful 3D coordinates while SpatialReasoner-Zero with 3D-aware process rewards produce good spatial reasoning trajectories.
> > >
> > > **About the claims in L260.** We would like to clarify that the arguments on L260 are directly supported by the results in Table 3 and the discussion in L257-258: RL-only model (SpatialReasoner-Zero) achieves 46.6% accuracy on unseen question types, outperforming SFT+RL (SpatialReasoner) with an accuracy of 43.4%. This observation is consistent with prior study comparing SFT and RL [12]. We agree with the reviewer that this is primarily an **empirical finding rather than a definitive conclusion**, and we are happy to soften the tone accordingly, *e.g.*, “Although SFT+RL improves over SFT, it does not fully match RL-only performance, suggesting that RL may encourage more flexible and better-generalizing reasoning than the more memorization-oriented SFT.”
> > >
> > > We hope this helps clarify any misunderstandings regarding the model implementation, and we would be happy to provide additional details or clarifications if the reviewer has any further questions or concerns.

---

### Official Review · Reviewer_Tsq1 · 2025-07-01

**Clarity:** 3
**Significance:** 3
**Originality:** 2
**Rating:** 5
**Confidence:** 4

**Summary:**

The paper aims to impart explicit 3D reasoning to VLMs. Specifically, the paper collects and pseudo-annotates images with 3D annotations and chains of thoughts, and uses this to fine-tune a model to make it 3D-aware. The paper explores doing this via, RL-only, SFT-only, SFT+RL and achieves good improvements over prior SOTA. The paper also ablates several design choices and studies failure modes of their models.

**Questions:**

I think this paper has good contributions and detailed results, and most of the weaknesses I listed are fixable. Hence, I am recommending acceptance for now, and looking forward to author rebuttal. The main actionable experiments in my opinion are:
-  SFT basleine with joint data from the proposed dataset instead of two-stage fine-tuning
- (can skip if this is painful / non-standard to setup) specialized models fine-tuned on the collected dataset

Besides these, fixing up the wtiting concerns that I pointed out in my review would help quite a lot!

**Ethical Concerns:**

["NO or VERY MINOR ethics concerns only"]

**Final Justification:**

The rebuttal addresses all my concerns. I will be recommending this paper for acceptance

**Limitations:**

yes

**Quality:**

3

**Strengths And Weaknesses:**

Strengths:
- The paper is well-written and easy to follow
- Making VLMs have explicit 3D reasoning is an interesting direction, and the paper achieves good results by doing so
- I liked the ablations in the papers, very thorough and cover a lot of important baselines. I also like the failure analysis on the CVD-bench datasets as well as analysis on which among the 3D perception and computation contributes to the most errors


Weaknesses:
- I strongly recommend moving the implementation details section to the main paper. In my first reading, I was lost on a) what is the base VLM the paper starts with b) what is the training data? What exactly is the SFT and RL model are trained on. These are crucial details to understand the experiments, and I think they need to be in the main paper
-  Spatialreasoner-zero is not defined and was misleading for me initially. I think this is following the deepseek’s terminology; however “-zero” to mean only RL is non-standard in the literature and I initially confused it with “zero-shot”. Please either introduce this term or instead use spatialreasoner-RL-only which is perhaps much more clear
- Another crucial detail from the implementation detail section was that the SFT model is trained on one set of dataset, and then an additional portion of dataset is used for RL fine-tuning. Naturally, one thinks of an SFT baseline which uses both the datasets and authors include it as SpatialReasoner-SFT (+ HQ SFT) [which was unclear to me before reading implementation details]. This baseline underperforms the model without the additional fine-tuning.
    - I think a more stable baseline would be to concatenate both datasets together and train an SFT baseline on them jointly instead of sequential training
    - In the sequential training, what checkpoint did the authors take? Since the second-stage SFT data is tiny, it is likely that a very early checkpoint would lead to the best performance, while the latter checkpoints would be quite overfitted [that’s why joint training would actually be more stable for getting a stronger SFT baseline. Unlike RL training, SFT training does not need two stages, and hence it should be allowed to see all the data together]
- The paper does not contain the details on how the pseudo-annotations were created. I would highly recommend the authors to add those details at least to the supplementary.
- Additional Baselines: Did the authors consider fine-tuning the “specialist” baselines like spatial-rgpt on their additional collected data?


I think this paper has good contributions and detailed results, and most of the weaknesses I listed are fixable. Hence, I am recommending acceptance for now, and looking forward to author rebuttal.

---

> ### Author Rebuttal · Authors · 2025-07-30
>
> We thank the reviewer for the time and effort to review our paper. We appreciate the positive feedback, mentioning that our paper is "well-written and easy to follow", our method of involving explicit 3D representations "interesting", and our experimental results "good and thorough". We discuss new results and address detailed concerns below. We will incorporate all valuable feedback and suggestions in our future revision.
>
> **Regarding implementation details of our SpatialReasoner and some writing concerns.** We thank the reviewer for raising the questions and will improve the clarity of the writing in our revision. We provide detailed responses as follows:
> 1. Our SpatialReasoner model is based on the Qwen2.5VL-7B model and post-trained on spatial VQA data generated with 3D pseudo-labels. We will introduce key implementation details at the beginning of the experimental section.
> 2. The training data of our SpatialReasoner in Table 1 and 2 includes: (1) 24k spatial VQA with pseudo 3D labels generated by our data synthesis pipeline, (2) another 1.2k spatial VQA synthesized and filtered by human verifiers, and (3) 24k standard VQA randomly sampled from the mix 665k SFT data used in LLaVA v1.5. For SFT, we jointly finetune SpatialReasoner on a combination of (1) + (2) + (3). For RL, we finetune the model on (2) only. **We apologize for the confusion, but for the main models in Table 1 and 2, there was only one SFT stage on combined data, and for ablation studies in L295-301 and Table 5, we consider a special SFT+HQ-SFT setting, which is trained on (1) + (2) + (3) and (2) sequentially, for fair comparison with the two-stage SFT+RL model.** Lastly we generate several variants of spatial-aware training data for ablation study, including spatial VQA data with only basic 3D perception and 3D computation QAs and spatial VQA data without reasoning trajectories or 3D representations.
>
> We appreciate the reviewer’s valuable feedback and will revise the paper to improve its clarity and presentation. In our revision, we will include key implementation details and will also open-source all code, data, and models to support reproducibility and benefit future research.
>
> **Regarding SpatialReasoner-zero.** The naming convention of our SpatialReasoner models follows the deepseek-series models. Specifically, SpatialReasoner-zero is finetuned with reinforcement learning without first-stage SFT. We agree with the reviewer that these naming conventions can be confusing and will introduce the design of each model clearly in the next revision.
>
> **Regarding the implementation details of our data generation pipeline.** Thank the reviewer for raising this issue. We will include more technical details in our revision and will open-source both our generated spatial VQA datasets and the code for the data synthesis pipeline to support reproducibility and benefit future research. We summarize our data generation pipeline below.
>
> To post-train multi-modal LLMs for better spatial understanding, previous works explored various visual foundation models such as Segment Anything and Depth Anything, and generated pseudo-labels for 3D depths, distances, and orientations [9,11,33] (L187-194). However, we extend beyond the previous spatial VQA data synthesis pipeline from the following two aspects:
> * Spatial relationships and reasoning trajectory: Compared to [9,11], which only studies object 3D depths and distances or [33] that studies a limited number of orientation relationships, we study a broader range of spatial relationships that are both challenging and require multi-step spatial reasoning. Answers in our generated spatial VQA data also contain chain-of-thought reasoning trajectories interleaved with explicit 3D locations and orientations, which are found to benefit models’ spatial reasoning.
> * Data quality: as we extend to more complex spatial relationships, we notice an increasing number of factual errors due to the long multi-step synthesis pipeline. Hence, we adopt a series of aggressive filtering based on model predictions and heuristics (L197-202), which is followed by an optional human verification stage. This leads to a smaller (24k vs over 1M) but much higher quality spatial VQA dataset. We further study the influence of data quality on SFT and RL in L278-284.
>
> **Finetuning a specialist model.** We agree with the reviewer that finetuning specialist models offers valuable insights into the nature of our synthetic data. However, as SpatialRGPT requires object masks as input, we instead finetune SpaceLLaVA, an open reproduction of SpatialVLM. Results in the table below show that finetuning on SpatialReasoner data leads to clear improvements on 3DSRBench for both LLaVA-v1.5 and SpaceLLaVA, highlighting the effectiveness of our data in enhancing the spatial reasoning abilities of multi-modal LLMs. Notably, the gap between the final models is small — this is because both models share similar architectures and the SpaceLLaVA training data contains some basic 2D spatial relationships (e.g., left and right in the 2D image plane) that are not very helpful for 3D spatial reasoning.
>
> | model | 3DSRBench |
> |:-:|:-:|
> | LLaVA-v1.5 | 38.1 |
> | LLaVA-v1.5 + our data | 44.2 |
> | SpaceLLaVA | 42.0 |
> | SpaceLLaVA + our data | 45.5 |

---

> > ### Comment · Reviewer_Tsq1 · 2025-08-03
> >
> > Perfect, this addresses all my concerns, thank you! I will be recommending this paper for acceptance

---

> > > ### Author Response · Authors · 2025-08-04
> > > **Response to Reviewer Tsq1**
> > >
> > > Thanks once again for your insightful comments and helpful suggestions. We will carefully address and incorporate them into our revision.

---

> > > ### Author Response · Authors · 2025-08-05
> > > **Response to Reviewer Tsq1**
> > >
> > > Dear Reviewer Tsq1, thank you again for the positive evaluation and constructive feedback. As we approach the end of the reviewer-author discussion period, we would like to kindly check if there are any new questions or concerns you would like us to address. If our rebuttal has fully addressed your concerns, we would be grateful if you might consider increasing your score to reflect a strong accept.

---

### Official Review · Reviewer_4UW2 · 2025-07-02

**Clarity:** 4
**Significance:** 3
**Originality:** 3
**Rating:** 5
**Confidence:** 4

**Summary:**

This paper proposes a LVLM called SpatialReasoner especially designed to solve 3D spatial reasoning problem based on explicit 3D representations. Besides, this work constructs the corresponding training data for SFT and RL training. The paper comprehensively analyzes the proposed model, including the effectiveness of explicit 3D representations and the multi-stage training strategy.

**Questions:**

1. I'm curious about the comparison between Qwen2.5 VL and SpatialReasoner when both are given ground truth 3D locations for reasoning to answer questions, thereby verifying whether SpatialReasoner actually learns something better after SFT and RL training.

**Ethical Concerns:**

["NO or VERY MINOR ethics concerns only"]

**Final Justification:**

Thanks for the detailed response which has addressed most of my concerns. Based the submission and the response, I am happy to raise my score to 5.

**Limitations:**

1. As the authors state, SpatialReasoner consistently performs explicit multi-step 3D reasoning, which is not efficient.
2. In more complex scenarios like multi-image inputs, it may be difficult for the model to generate such explicit 3D representations.

**Paper Formatting Concerns:**

N/A.

**Quality:**

3

**Strengths And Weaknesses:**

Strength:
1. The paper is well-written.
2. The idea of reasoning with explicit 3D representations is reasonable, and the comparison with Gemini 2.0's reasoning is especially convincing regarding its necessity.
3. The experiments on RL training are interesting and reveal some intriguing findings.
4. The open access commitment of this work, especially the RL training code and data, will be beneficial to the community.


Weakness:
1. The performance drops on CVBench-3D distance questions remains confusing, though the authors claim this can be attributed to shortcut-driven 2D biases in CVBench-3D. In my opinion, reasoning in 3D should not perform worse in such simpler scenarios. I think this can be attributed to errors in predicted 3D positions. I would like to see whether the model can outperform Qwen2.5 VL given ground truth 3D positions.
2. The reward design for RL training is ambiguous, especially for the 3D-aware process reward. It would be better to express it with a clearer formula.

---

> ### Author Rebuttal · Authors · 2025-07-30
>
> We thank the reviewer for the time and effort to review our paper. We appreciate the positive feedback, mentioning that our paper is "well-written", our method of involving explicit 3D representations "reasonable and convincing", and our experiments "interesting and revealing intriguing findings". We discuss new results and address detailed concerns below. We will incorporate all valuable feedback and suggestions in our future revision.
>
> **Inference with ground-truth 3D locations.** We agree with the reviewer that inferencing Qwen2.5-VL and our SpatialReasoner with groundtruth 3D information provides valuable insights into the models. Therefore, we obtain the ground-truth 3D object locations from the Omni3D dataset and append the information to the end of the question as part of the prompt.
> * Results show that with 3D ground-truths, our SpatialReasoner achieves an almost perfect performance of 97.9% while Qwen2.5VL performs similarly to the standard evaluation. This demonstrates that Qwen2.5VL does not rely on genuine 3D spatial reasoning to answer depth- and distance-related questions. It is likely making predictions by exploiting 2D shortcuts. In contrast, our SpatialReasoner learns to ground objects in 3D space and perform reliable spatial reasoning on 3D information.
> * Moreover, these results suggest that shortcuts in CV-Bench-3D can lead to deceptively strong performance, undermining meaningful evaluation. Truly solving the spatial reasoning challenges in CV-Bench-3D requires both a robust visual module for accurate 3D parsing and a powerful reasoning model capable of genuine spatial understanding.
>
> | evaluation | model | CV-Bench-3D | depth | distance |
> |:-|:-|:-:|:-:|:-:|
> | standard | Qwen2.5VL | 82.8 | 82.5 | 83.2 |
> | standard | SpatialReasoner | 80.3 | 87.3 | 73.3 |
> | with 3D GT | Qwen2.5VL | 82.5 | 83.3 | 82.8 |
> | with 3D GT | SpatialReasoner | 97.9 | 99.5 | 96.3 |
>
> **Clarification on the reward designs of reinforcement learning.** We thank the reviewer for raising this concern. We introduce the motivation and design of our reward functions as follows, and will rewrite this section with detailed formulas and examples in our revision.
>
> In this work, we consider three types of reward functions: accuracy reward, format reward, and 3D-aware process reward.
> * The accuracy and format rewards follow the DeepSeek-R1 design that checks if the predicted final answer matches the ground truth and if the format of the answer follows a reasoning step \<think\>...\</think\> and an answering step \<answer\>...\</answer\>.
> * Process rewards have been considered in [R1,R2,R3]. We propose novel 3D-aware process rewards to encourage the model to predict necessary 3D representations in the reasoning step. This is realized by implementing a range of regex patterns that recognize spatial representations --- 3D locations, distances, front/left direction, angles, etc. Then, given the question types, we reward the model if the reasoning trajectory contains relevant spatial representations. As an example, for a multi-object distance question, we give reward=1 if the reasoning trajectory contains 3D locations and distances, and reward=0 if the reasoning trajectory contains 3D orientations and directions.
>
> | reward | value | example |
> |:-|:-:|:-|
> | accuracy | 1 | \<answer\>red sofa\</answer\> (*matches the ground truth*) |
> | | 0 | \<answer\>blue sofa\</answer\> (*does not match the ground truth*) |
> | process | 1 | \<think\>The 3D location of the red sofa is…\</think\> \<answer\>red sofa\</answer\> |
> | | 0 | \<think\>The 3D location of the red sofa is… red sofa\</answer\> |
> | 3D-aware process | 1 | \<think\>The 3D location of the red sofa is…\</think\> |
> | | 0 | \<think\>The red sofa is next to the dining table…\</think\> |
>
> **Extension to multi-image inputs.** We agree with the reviewer that extending our SpatialReasoner to multi-image or video inputs presents nontrivial challenges. However, as motivated by the comparison between Gemini 2.0 and SpatialReasoner in Figure 1, we argue that adopting explicit 3D representations remains essential for MLLMs to achieve reliable and accurate spatial reasoning. While multi-view inputs introduce complexities such as varying camera viewpoints, recent feed-forward transformer, such as DUSt3R [R4] and VGGT [R5], have shown promising results in predicting grounded geometries (e.g., point maps) within the 3D coordinates defined by the first frame. Although some technical issues remain unexplored, we are optimistic about the potential of our explicit 3D representations to generalize to these more complex settings.
>
> [R1] Scaling LLM Test-Time Compute Optimally can be More Effective than Scaling Model Parameters
>
> [R2] Math-Shepherd: Verify and Reinforce LLMs Step-by-step without Human Annotations
>
> [R3] VisualPRM: An Effective Process Reward Model for Multimodal Reasoning
>
> [R4] DUSt3R: Geometric 3D Vision Made Easy
>
> [R5] VGGT: Visual Geometry Grounded Transformer

---

> > ### Author Response · Authors · 2025-08-05
> > **Response to Reviewer**
> >
> > Dear Reviewer 4UW2, thank you for your time and effort to review our paper. We appreciate the constructive feedback and will integrate them in our revision. As we approach the end of the reviewer–author discussion period, we would like to kindly check if our rebuttal has addressed your concerns or if there are any follow-up questions we can clarify. If our rebuttal has fully resolved your concerns, we would be grateful if you might consider increasing your rating.

---

> > ### Comment · Reviewer_4UW2 · 2025-08-06
> > **On the response**
> >
> > Thanks for the detailed response which has addressed most of my concerns. Based the submission and the response, I am happy to raise my score to 5.

---

> > > ### Author Response · Authors · 2025-08-06
> > > **Response to Reviewer 4UW2**
> > >
> > > Thank the reviewer again for the insightful comments and helpful suggestions. We will carefully address and incorporate them into our revision.

---

### Comment · Area_Chair_ENmz · 2025-08-04
**[Action Required] Participate in Reviewer-Author Discussion**

Dear Reviewers,

As the author-reviewer discussion period will end soon (Aug 6, 11:59 PM AoE), please take a moment to read the authors’ responses and post a reply - either to acknowledge their clarifications or to raise any remaining concerns.

Thank you for your time and contributions to the review process.

Best regards,

AC

---

### Decision · Program_Chairs · 2025-09-17

**Decision:**

Accept (poster)

**Comment:**

This paper presents SpatialReasoner, a large vision-language model designed for 3D spatial reasoning with explicit 3D representations and a multi-stage training framework combining supervised fine-tuning and reinforcement learning.

Three reviewers (4UW2, Tsq1, and X79Q) rated the paper as "accept" and acknowledged its novelty, clear presentation, and thorough experiments. One reviewer (rCGV) provided a borderline reject score. While acknowledging the novelty, reviewer rCGV remained unconvinced that the models were genuinely learning accurate 3D representations and had concerns about the claim in L260.

Overall, after considering all reviews, rebuttals, and discussions, the AC finds the paper is timely and relevant given growing interest in integrating spatial reasoning into multimodal foundation models. Given the clear majority of positive reviews and the author rebuttal's success in addressing most concerns, the AC recommends acceptance.  The authors are recommended to revise the statement in L260 and expand their discussion to provide a deeper analysis of the experimental results.